# Helminths of the Wild Boar (*Sus scrofa*) from Units of Conservation Management and Sustainable Use of Wildlife Installed in the Eastern Economic Region of Mexico

**DOI:** 10.3390/ani11010098

**Published:** 2021-01-06

**Authors:** Jorge-Luis de-la-Rosa-Arana, Jesús-Benjamín Ponce-Noguez, Nydia-Edith Reyes-Rodríguez, Vicente Vega-Sánchez, Andrea-Paloma Zepeda-Velázquez, Víctor-Manuel Martínez-Juárez, Fabián-Ricardo Gómez-De-Anda

**Affiliations:** 1Immunoparasitology Laboratory, Institute for Epidemiological Diagnosis and Reference, Ministry of Health, México City 01480, Mexico; delarosa.jorgeluis@yahoo.com; 2Academic Area of Veterinary Medicine and Zootechnics, Institute of Agricultural Sciences, Autonomous University of the State of Hidalgo, Tulancingo 43600, Hidalgo, Mexico; mvzjesusponce@gmail.com (J.-B.P.-N.); nydia_reyes@uaeh.edu.mx (N.-E.R.-R.); vicente_vega11156@uaeh.edu.mx (V.V.-S.); andrea_zepeda@uaeh.edu.mx (A.-P.Z.-V.); victormj@uaeh.edu.mx (V.-M.M.-J.)

**Keywords:** wild boar, feral pig, parasite, helminth, antibody, Mexico

## Abstract

**Simple Summary:**

Wild boars (*Sus scrofa*) were introduced in Mexico for sport hunting and meat trading for human consumption, but their role in the transmission of diseases to human or domestic animals is limited. Thus, we did research looking for parasitic worms in wild boars that live in three units of conservation management and sustainable use of wildlife installed in the eastern economic region of Mexico. Samples of feces and serum were collected from 90 animals that came from three different ranches. Stool examination and antibody determination were performed. Eggs of *Strongyloides* sp. (72.2%), *Metastrongylus* sp. (57.7%), *Oesophagostomum* sp. (53.3%), and *Trichuris* sp. (37.7%) were found in addition to oocysts of *Eimeria* sp. (75.6%). Antibodies to *Fasciola* (8.9%), *Taenia* (4.4%), *Ascaris* (32.2%), *Toxocara* (20%), and *Trichinella* (5.5%) were found. This is the first report of parasitic worms of wild boar produced in Mexico. The importance of the results is based on the limited data available about the impact of wild boar and feral pigs on the transmission of diseases to domestic animals. This study identifies the potential risk of wild boar as a transmission channel of diseases than can have an impact on public health.

**Abstract:**

Wild boars (*Sus scrofa*) were introduced in Mexico for sport hunting and meat trading for human consumption, but the available data regarding their role in pathogen transmission are limited. This research and field work aimed to identify the helminths of the wild boar produced in three units of conservation management and sustainable use of wildlife placed in the eastern economic region of Mexico. Samples of feces and serum were collected from 90 animals that came from three different ranches. Stool examination and antibody determination to *Fasciola hepatica*, *Taenia crassciceps*, *Ascaris suum*, *Toxocara canis* (ELISA), and *Trichinella spiralis* (Western blot) were performed. In addition, 30 diaphragm samples from one ranch were obtained for artificial digestion. Eggs of *Strongyloides* sp. (72.2%), *Metastrongylus* sp. (57.7%), *Oesophagostomum* sp. (53.3%), and *Trichuris* sp. (37.7%) were found in addition to oocysts of *Eimeria* sp. (75.6%). Antibodies to *Fasciola* (8.9%), *Taenia* (4.4%), *Ascaris* (32.2%), *Toxocara* (20%), and *Trichinella* (5.5%) were found. The eggs of *Strongyloides* and *Oesophagostomum* were associated to female hosts. One nematode larva was found by artificial digestion. This is the first report to identify helminths from wild boars in Mexico. In addition, this study identifies the potential risk of the wild boar as a transmission channel of parasites that can have an impact on public health.

## 1. Introduction

Wild boar (*Sus scrofa*) is a wildlife animal highly used for food and sport hunting all over the world [1]; however, hunting without appropriate hygiene measures during the handling of meat and offal can become a potential risk for pathogen transmission to humans and other domestic animals. In Mexico, hunting can only be carried out legally in units for the conservation management and sustainable use of wildlife (UMA, according to the acronym in spanish for animal management units) that are registered according to the corresponding local regulations [2]. In México, during 2017, a total of 1722 UMA were registered with 38.5 million hectares, which is equivalent to 19% of the national territory [3].

Wild boar and feral pig (both of genus *Sus scrofa*) are considered invasive animals, due to their fast reproduction rate (on average three to five cubs per litter and two litters per year), and the absence of predators influences the habitat modification and the competition with regional animal species. In many countries of the world, the populations of feral pigs, wild boars, and hybrids are continuously studied to identify the risk of pathogen transmission [4,5,6]. Wild boars hosts different stages of development of helminths that have a great importance on public health and veterinary topics. To mention an example, wild boards and swines, in general, host the adult worms of *Ascaris suum*, and larvae of *Taenia solium*, *Echinococcus granulosus*, and *Trichinella spiralis*; indeed, wild boards can act as a paratenic and accidental host of *Toxocara* and *Fasciola*, respectively [7,8]. In addition, wild boar meat is sought as a nutrients source because wild boars have a greater loin area and greater amount of fat compared to domestic pigs; indeed, between wild boars and domestic pigs, the variance in the composition of fatty acids contributes to differentiate taste in the cooked meat [1].

Nowadays, meat demand from non-conventional animals, including wild boar, has increased worldwide, which has motivated the implementation of food safety measures and the intentional search for pathogens in products of animal origin [8,9]. There are several methods to search for helminths, such as the artificial digestion of meat to determine the presence of tissue nematodes such as *Trichinella spiralis* [10]. Additionally, there are indirect methods, such as the enzyme-linked immunosorbent assays (ELISA) and Western blot, which are valuable to identify the presence of antibodies. As a consequence, antibodies are useful to evaluate the circulation of infectious agents among the populations [11,12]. Thus, given the need to identify the potential risk of wild boar (*Sus scrofa*) from units of conservation management and the sustainable use of wildlife installed in the eastern economic region of Mexico as a transmitter of diseases, this research and field work aimed to search for helminths with high relevance in public health.

## 2. Materials and Methods

### 2.1. Ranches for the Conservation Management and Sustainable Use of Wildlife

Wildlife from ranches for the conservation, management, and sustainable use of wildlife were studied. This field work has been placed in three ranches in the eastern region of Mexico located in the Federal Entities of Hidalgo (Mineral de la Reforma Municipality, 20°2′44.68″ N, 98°43′15.88″ W), Puebla (San Bernardino Tlaxcalancingo town, 19°1′44″ N, 98°16′30″ W), and Tlaxcala (Municipality of Calpulapan 19°35′00″ N, 98°34′00″ W). Different studies on the human development index consider that Hidalgo, Puebla, and Tlaxcala are the most rural entities in the country [13]. This kind of condition is related to the life cycle of parasites. 

### 2.2. Biological Samples

The biological samples (feces and blood) were collected by sampling convenience. Samples were collected during the process implemented by each ranch to examine their animals. Briefly, between the months of February and June, potatoes (*Solanum tuberosum*) are planted randomly in open areas of the ranch as supplement for wild boars. In the months that potatoes are not planted, the animals are attracted to sacks of potatoes that are nearly expired. In Hidalgo’s ranch, samples of 23 females of wild boars and 7 males were obtained; in Puebla, samples were obtained from 20 females and 10 males; while in Tlaxcala, samples of 18 females and of 12 males were obtained. A total of 90 samples were obtained (61 from females and 29 from males). The stool samples were taken immediately upon deposition and stored in plastic bags that were labeled with the animal’s identification. The blood samples were obtained by the personnel of each ranch; 3 to 5 mL of blood were collected from the jugular vein and then, the samples were transported at 4 °C to separate the clot in the laboratory centrifugations at 3500 rpm for five minutes to obtain the serum. Serum samples were packaged in 1.5 mL tubes and frozen at −20 °C until use. Additionally, from the ranch of Tlaxcala, 30 diaphragm samples, each of 50 g, were obtained by hunters’ donations.

### 2.3. Stool Examination

Stool samples were examined by the float concentration technique with saturated saline solution. Four grams of feces were homogenized in 56 mL of tap water and allowed to stand for 30 min. Then, samples were sieved, and 10 mL of the suspension were taken to centrifuge for 10 min at 500× *g*. The pellet was homogenized with 4 mL of saturated saline solution; then, a McMaster-type counting chamber was filled and allowed to stand for 5 min. All the eggs within the grids of each chamber were counted to determine the parasites per gram of feces; the eggs outside the boxes were not considered. The total of eggs from the two chambers was added and multiplied by fifty [14]. 

### 2.4. Artificial Digestion

Meat samples were analyzed by artificial digestion according to the procedure and recommendations previously described [15,16]; digestion fluid was prepared with 0.5% pepsin (Sigma-Aldrich, St. Louis, MO, USA) in 0.2% hydrochloric acid and incubated for 30 min at 44 °C. The search for larvae of helminths was carried out at 10× magnification in bright field microscopy.

### 2.5. Antigens

The antigens of *Ascaris suum* and *Fasciola hepatica* were total extracts from adult worms obtained from natural infections. *Toxocara canis* antigens were excretion and secretion products of adult worms obtained from natural infection. In the case of *Trichinella spiralis*, the excretion and secretion products of the muscle larvae, strain MSUS/ME/92/CM-92, were prepared as previously reported [17]. Briefly, the larvae were isolated by the artificial digestion of meat obtained from rats experimentally infected and then incubated in RPMI 1640 medium (Gibco BRL, Grand Island, NY, USA) at 37 °C for 48 h in a humid atmosphere of 95% air and 5% CO_2_. In the case of *Taenia crassiceps* WFU strain, the antigen obtained was the vesicular fluid recovered from the metacestode. The strains of *Trichinella* and *Taenia* have been maintained by serial passages in mice at the Institute for Diagnostic and Epidemiological Reference (InDRE), Ministry of Health, Mexico. The Immunological Research Coordination (InDRE) according to the Mexican regulations (Norma Oficial Mexicana NOM-062-ZOO-1999, technical specifications of proper care and use of laboratory animals in research, testing, teaching, and production) approved the procedure of the experimental infection to obtain the antigens.

### 2.6. Antibody Determination 

Antibodies to *Ascaris*, *Taenia*, *Toxocara*, and *Fasciola* were determined by ELISA, while antibodies to *Trichinella* were determined by Western blot. The ELISA procedure was carried out as previously described [18], using an antigenic solution (0.003 mg/mL) to coat high-binding polystyrene plates (Corning, Tewksbury, MA, USA). The serum sample was used by triplicate at 1:100 dilution and an anti-pig IgG (whole molecule) peroxidase antibody produced in rabbit (Sigma-Aldrich Corp., St. Louis, MO, USA) was used at 1:4000 dilution. Reaction was revealed with a mixture solution prepared with o-phenylenediamine (Sigma-Aldrich, St. Louis, MO, USA). Absorbance values were determined with an ELISA reader (Boehringer, Mannheim, Germany) at 490 nm. The cutoff point was calculated for each antigen using absorbance average values plus three standard deviations of 13 normal pig sera. The antibody detection to *Trichinella* was carried out by Western blot, as previously reported [17]. Briefly, 300 µg of antigen were separated under reducing conditions on an 11% SDS-PAGE mini-gel (equivalent to 4 μg/mm of antigen). Afterward, the antigen was transferred to a nitrocellulose membrane using a vertical electro-transfer system semi-dry (Bio-Rad, Hercules, CA, USA). Strips of 0.3 cm wide were incubated with serum samples diluted 1:800 and then incubated with a rabbit anti-pig IgG peroxidase conjugate (Sigma-Aldrich) diluted 1:2000. The reaction was developed with a diaminobenzidine substrate–chromogen solution. Serum samples that were reactive with the specific diagnostic antigens of 45, 49, and 53 kDa were classified as positive.

### 2.7. Statistical Analysis

The data obtained were analyzed as a simple frequency. To determine the association of the presence of parasites and the sex of the wild boar as well as the presence of parasites and the ranch where the stool samples were collected, 2 × 2 contingency tables were made and analyzed with the mid-p exact test by means of the OpenEpi: Open-Source Epidemiologic Statistics for Public Health, version 3.01 (https://www.openepi.com/). A serum sample was considered positive in the antibody determination by ELISA (antigens of *Fasciola hepatica*, *Taenia crassiceps*, *Ascaris suum* and *Toxocara canis*) if the absorbance value was greater than the cutoff point (previously calculated with negative samples). In the case of antibodies against *Trichinella spiralis* antigens, a sample was considered positive if the diagnostic antigens of 45, 49, and 53 kD were recognized in the Western blot.

## 3. Results

### 3.1. Determination of Gastroenteric Parasites

Table 1 shows the results of stool examination. In general, 76/90 (84.4%) of analyzed samples showed parasitic stages. Concomitant infections of at least two parasites were observed in 56/90 (62.2%) samples. The identified helminth eggs were *Strongyloides* sp. in 65/90 (72.2%) samples, *Metastrongylus* sp. in 52/90 (57.7%) samples, *Oesophagostomum* sp. in 46/90 (53.3%) samples, and *Trichuris* sp. in 34/90 (37.7%) samples; additionally, oocysts of the apicomplex protozoan *Eimeria* sp. were observed in 68/90 (75.6%) samples (Figure 1, panels A, B, C, D, and E). During the overall analysis of data obtained with the 90 stool samples, we observed that *Strongyloides* and *Oesophagostomum* eggs were associated with the female sex of wild boar (*p* < 0.05; Mid-P exact); however, during the analysis by the ranch, this association only held for *Strongyloides* at Hidalgo ranch. In contrast, in the same Hidalgo ranch, *Oesophagostomum* was more frequent in males than in females (*p* < 0.05; Mid-P exact). 

### 3.2. Artificial Digestion

A nematode larva (Figure 1, panel F) was recovered from the digestion of the 30 diaphragm samples (ranch from Tlaxcala). The larva that was recovered measured 1280 μm long by 35.16 μm wide in the front part and 21.68 μm wide in the back part, with a cuticle 2.01 μm wide (Figure 1, panels G and H); however, in the author’s opinion, there is not enough evidence to determine their genus. The found larvae were compared with the muscular larvae of *Trichinella* recovered from experimental infected mice. The measurements of a *Trichinella* larva were 1254.27 ± 80.23 μm in length by 31.89 ± 3.22 μm in width in the anterior part and 21.08 ± 1.33 μm in the posterior part; the cuticle was 1.54 ± 0.074 μm wide. The stichosome measures 456.65 ± 29.59 μm long and 24.74 ± 2.6 μm wide.

### 3.3. Antibodies

Table 2 shows the distribution of wild boar serum samples that were reactive by ELISA to helminth antigenic extracts; the absorbance value of each sample is shown in Figure 2, and the Western blot for the determination of antibodies to *Trichinella* is shown in Figure 3. In general, a positivity frequency of 8.9% was found for *Fasciola* antigens, while 4.4% was found for *Taenia*, 32.2% was found for *Ascaris*, and 20% was found for *Toxocara*. In the case of *Trichinella* antigens, five samples (5.5%) were reactive to the bands of 45, 49, and 55 kDa. The recognition of these bands is considered the immunological diagnostic criterion for the diagnosis of *Trichinella spiralis* [11,18]. During the overall analysis of data, we observed that antibodies to *Toxocara* were associated with the female sex (*p* < 0.05; Mid-P exact) at the ranch located in Hidalgo. 

## 4. Discussion

The wild boar was introduced in Mexico for hunting, providing an economic improvement in areas where it lives. However, in the absence of data on the parasitic fauna of the Mexican wild boar, it is unknown if there is transmission to humans, pigs, and other domestic animals. On this research and field work, we searched for helminths hosted by wild boars from units of conservation management and sustainable use of wildlife installed in the eastern economic region of Mexico. The data presented on this paper is the first to be reported in Mexico for wild boars and even for feral pigs. The hunting practice in Mexico is an industry that could generate important economic, social, and environmental benefits if it is properly handled [2]. However, additional studies are needed to determine the impact of meat consumption on hunting practice, including the disposal of the carcass [19]. Although in each ranch, the feeding is guaranteed for animals, the instincts cannot be avoided. Thus, wild boar and feral pig can feed on other animals or on their corpses, favoring the life cycles of parasites such as *Trichinella*. In this way, wildlife production systems can generate synanthropic cycles, where contact between domestic and wild animals can promote the pathogen transmission [20].

The evaluation of wild boar and feral pig populations is a common practice in some countries to identify the impact of animals during their introduction into ecosystems; indeed, the risk of pathogen transmission to humans and domestic animals is also evaluated [4,5,6]. In Mexico, feral animals of genus *Sus scrofa* are considered habitat invaders, and there is a concern about the impact they may have on native plant and animal species. Several studies have been carried out in Mexico to determine the population size and the distribution of the feral pig in the country [21,22,23]. Feral pigs can be distributed in areas with natural vegetation and in deforested areas such as crops and induced grasslands [24]; however, in Mexico, studies on the parasitic fauna of wild boars and feral pigs are still insufficient. At the moment, it is known that wild boars in Mexico are limited to hunting ranches; however, it is unknown how many of them have become feral animals. Additionally, the number of domestic pigs that have become feral animals remains unknown, and much less is known about the population of pig–wild boar hybrids. There have been several attempts at the national level to determine the number of these swine in the country. In the states from the northern and central economic regions of Mexico, swine are considered invasive species. Nowadays, there are no systematic data about the prevalence and distribution of wild boars, which are also influenced by eating habits, forced migrations, and human activity, the latter seeing these animals as an opportunity for profit, simply by hunting them without registration of health or economic activity. In this way, it is desirable that the real size of the swine feral animal population be known and its impact as an invasive species and transmitter-reservoir of pathogens verified.

Arnaud-Franco et al. [25] studied the ectoparasites of the feral pig from Northwest Mexico, registering a moderate infection by lice with an abundant presence of eggs in the hair; additionally, the authors registered an individual affected by scabies. In another study, Pérez-Rivera et al. [26] recorded the presence of antibodies against the swine influenza virus (30.7%), *Leptospira* (25.7%), *Salmonella* (28.7%), and *Brucella* (14.2%) in 70 feral pigs of the northwest economic region of Mexico.

In other studies, carried out in different parts of the world, such as Spain and Argentina, to name a few, different helminths have been found. For example, in the province of Valencia (Spain), during necropsy (*n* = 47), the following helminths were found [27]: *Taenia hydatigena* (cysticercus) (19%), *Ascarops strongylina* (87%), *Physocephalus sexalatus* (6%), *Ascaris suum* (2%), *Metastrongylus* spp. (85%), *Capillaria* sp. (2%), and *Macracanthorhynchus hirudinaceus* (21%). In Samborombón Bay (Argentina), during the autopsy and stool examination (*n* = 30), the following helminths were reported [28]: *Macracanthorhynchus hirudinaceus* (33%), *Ascaris suum* (22%), *Oesophagostomum dentatum* (4%), *Globocephalus* sp. (7.5%), *Metastrongylus* sp. (7.5%), *Hyostrongylus* sp. (18.5%), *Trichuris* sp. (4%), and the larvae of *Echinococcus* sp. (20%). Recently, it was documented that wild boar can host at least 11 genera of nematodes, 1 of acanthocephalans, 3 of cestodes, and 2 of trematodes [29]. In this way, the wild boar is a host that allows the growth of many species of helminths. Thus, the wild boar becomes a potential transmission channel for parasites to humans or domestic animals, without considering the capacity of the wild boar as a reservoir for a great biodiversity of helminths. However, the biodiversity of helminths in wild boars and feral pigs is associated with the presence of abiotic factors, such as constant humidity and the absence of extreme temperatures that, combined with the presence of vectors and other hosts, can drive helminths to conclude its life cycle. Additionally, it has been discussed whether there are strains that are more susceptible or resistant to parasites. Although the topic is still under discussion, it is known that the success of a parasite to establish itself in a host depends largely on the ability of the host itself to mount an adequate immune response. This issue can become controversial, because much has been discussed regarding whether the immune response associated with helminths kills them or helps them to survive [30,31]. Results here reported agree with data previously shown by other authors; however, we found in general that female hosts were associated with *Strongyloides* and *Oesophagostomum* eggs. In contrast, in the same Hidalgo ranch, *Oesophagostomum* was more frequent in males than in females. These data only confirm that is hard to define whether the host sex has any influence on the establishment of parasites and therefore, studies on this topic must continue. There are some examples of host sex influence in the development of the parasite; for example, in *Taenia crassciceps*, the parasite strain is propagated in female mice to ensure the success of the procedure [32]; but in other cases, as in human trichomoniosis, the female host is marked associated with clinical development.

In this study, we did not find *Ascaris*, *Toxocara*, or any cestode eggs in the feces of the wild boar, but we did find IgG antibodies. This may be associated to the fact that the detection of antibodies in the absence of clinical data is not indicative of active infection but rather of contact with the etiological agent. It is also known that the oviposition of helminths is not a continuous event, but rather, it is intermittent. Consequently, the determination of antibodies and the stool examination techniques should be considered as complementary to make a global interpretation. There are factors, such as the sensitivity of stool examination techniques and the presence of cross-reacting antibodies in the different groups of helminths, which, if not properly interpreted, could over- or underestimate the prevalence of etiological agents [33]. In fact, in this study, we take advantage of the cross-rection antibodies that exist between the cestodes to use the vesicular fluid of *Taenia crassiceps* to estimate the seroprevalence to these helminths in wild boar. Thus, antibody data of *Taenia crassiceps* have a particular interest, since they are indirect evidence of *Taenia solium* or *T. hydatigena* presence or even any other cestode. The presence of cross-reactive antibodies between cestodes have previously been demonstrated [34]. 

On the other hand, what is remarkable in the previous reports we cited is the nonappearance of *Strongyloides* and *Trichinella*. The meat inspection of wild boar and pork are of special importance to detect muscle larvae of *Trichinella*. The nematode can be transmitted to humans by the ingestion of raw or undercooked meat that harbors viable larvae. In fact, numerous epidemic outbreaks of human trichinellosis have been recorded throughout the world associated with the consumption of wild boar meat [1]. Recently, Franssen et al. [35] published the results obtained with a microbial risk assessment model that simulates the transmission of *Trichinella* from wildlife toward domestic one. This model estimated that in Poland, the incidence of human trichinellosis due to the consumption of pork produced in uncontrolled stables is 0.90 cases per million people per year, while the incidence due to the consumption of wild boar meat is 1.97. Rostami et al. [36] reported a meta-analysis of 21 studies involving 16,327 wild boars from 15 countries and found that the global seroprevalence is 6%, although by continent, the estimated seroprevalence was 9% for North America, 7% for Europe, and the data reported in Asia and Oceania are 3% in both areas. Here, we report an antibody-positivity frequency of 5.5% (Table 2).

In Mexico, the search for parasites in the carcasses of domestic horses and pigs is done in controlled slaughterhouses. However, there are no regulations for the sanitary verification of wild game. Furthermore, knowledge about the interaction between wildlife and domestic animals is insufficient; so, the study of the etiological agents in wild animals should be a constant to know the risk of transmission of pathogens. Considering that trichinellosis is a zoonosis that is transmitted by the consumption of pig meat with viable muscular *Trichinella* larvae, in this work, we carried out an intentional search for these larvae by the artificial digestion of the diaphragm samples. If we had found them, we would have tried to propagate the life cycle in an experimental way to later determine genus and species. However, we only found one larva, so we proceeded to morphologically compare the recovered larvae with those of *Trichinella* that we propagated in the laboratory. We know that *Trichinella* larvae are not the only ones that can establish themselves in skeletal muscle; other larvae, such as those of *Toxocara* or *Baylisascaris*, could also settle in skeletal muscle. Therefore, since only one larva was found in the artificial digestion, it was a limitation to be able to make its classification. This finding suggests that the prevalence of tissue nematodes is low and therefore opens a possibility for future epidemiological studies.

## 5. Conclusions

In conclusion, this is the first report that identifies the wild boar helminths of Mexico. Considering that the demand for wild boar meat consumption in Mexico is underestimated, more studies should be carried out to determine the impact in the transmission of these parasites to humans and animals destined for consumption and, if pertinent, further studies should reinforce sanitary regulations. Finally, in this work, the potential risk of wild boar as a transmission channel of parasites with importance for public health is identified.

## Figures and Tables

**Figure 1 animals-11-00098-f001:**
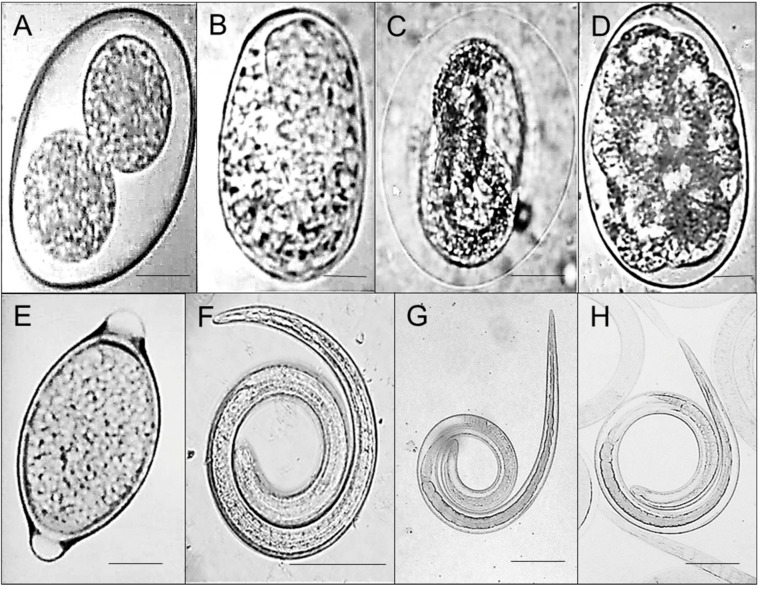
Parasites found during the analysis of biological samples from wild boars from units of conservation management and sustainable use of wildlife installed at the eastern economic region of Mexico. Results of stool exam showed cysts of *Eimeria* sp. ((**A**), the bottom right line measures 5 um) and eggs of *Strongyloides* sp., (**B**), *Metastrongylus* sp. (**C**), *Oesophagostomum* sp. (**D**) and *Trichuris* sp. (**E**); the bottom right line measures 10 μm. (**F**) shows the larvae of the nematode recovered by enzymatic digestion and (**G**,**H**) show a *Trichinella* muscle larvae obtained from the enzymatic digestion of an experimentally infected mouse; lines are 100 μm.

**Figure 2 animals-11-00098-f002:**
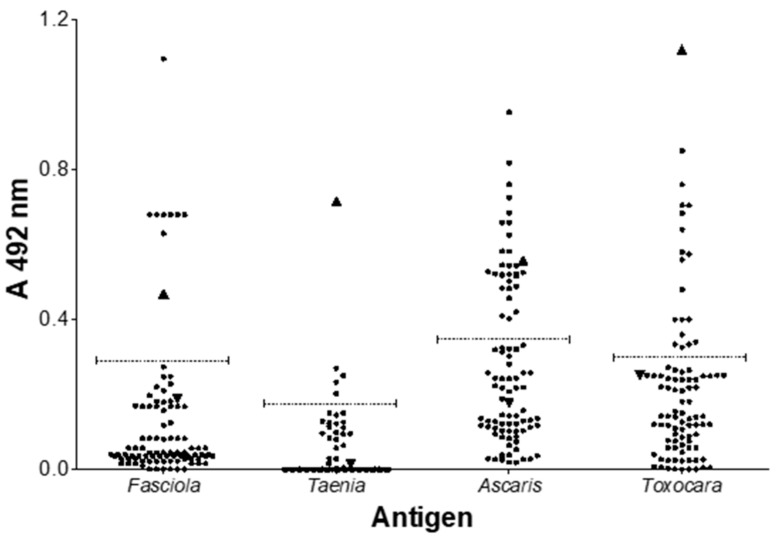
Reactivity of wild boar serum samples against different antigenic extracts of helminths. The absorbance values obtained by ELISA are shown as a function of the antigens of *Ascaris suum*, *Toxocara canis*, *Fasciola hepatica,* and *Taenia crassiceps*. The dotted lines represent the cutoff value. The negative control (▼) is a pig serum obtained from a technical farm. The sera that were used as positive controls (▲) in the *Ascaris* (pig), *Toxocara* (dog), and *Fasciola* (cow) tests were obtained from natural infections, while in the case of *T. crassiceps*, the serum of an experimentally infected mouse was used.

**Figure 3 animals-11-00098-f003:**
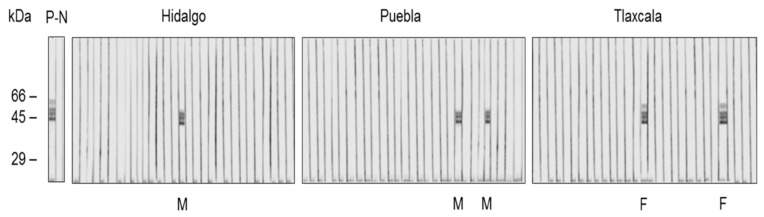
Western blot reactivity of the wild boar serum samples against the excretion and secretion antigens of *Trichinella spiralis*. The figure shows the reactivity of each serum sample obtained from the wild boars. The strips are grouped by location (Hidalgo, Tlaxcala, Puebla). The positive control serum (P) corresponds to an experimentally infected rat serum, while the negative control serum (N) was from healthy rat. The molecular weight marker is shown to the left of the photo. The positive samples corresponded to three male wild boars (M) and two females (F).

**Table 1 animals-11-00098-t001:** Gastroenteric parasites found in wild boars from the eastern economic region of Mexico.

Helminth	Positive (%)	F/M ^1^ (*p*) ^2^	Number of Detected EPG/OPG ^4^	Frequency for Each Ranch ^3^		
				Tlaxcala	Puebla	Hidalgo
				F/M (*p*)	F/M (*p*)	F/M (*p*)
*Strongyloides*	65 (72.2)	39/26 (0.009902)	650 ± 321	12/8 (0.9)	17/9 (0.8)	10/7 (0.005516)
*Metastrongylus*	52 (57.7)	33/19 (0.3)	730 ± 425	13/7 (0.5)	9/6 (0.5)	11/6 (0.1)
*Oesophagostomum*	46 (53.3)	25/21 (0.005953)	540 ± 152	7/7 (0.3)	13/7 (0.8)	5/7 (0.0001608)
*Trichuris*	34 (37.7)	20/14 (0.2)	270 ± 131	5/5 (0.5)	6/7 (0.05)	9/2 (0.7)
*Eimeria*	68 (75.6)	44/24 (0.3)	710 ± 294	15/10 (0.9)	12/7 (0.9)	17/7 (0.1)

^1^ Female/Male. ^2^ Value of the statistical probability calculated with Mid-P exact. ^3^ A total of 30 samples were collected in each ranch (unit of conservation management and sustainable use of wildlife). ^4^ Eggs of helminths or oocysts of *Eimeria* per gram of feces (EPG/OPG).

**Table 2 animals-11-00098-t002:** Distribution of ELISA positive serum samples of wild boars to helminth antigenic extracts.

		Positive Samples				
		*Fasciola*	*Taenia*	*Ascaris*	*Toxocara*	*Trichinella*
Ranch Sex ^1^ F/M		F (%) M (%) *p* ^2^ CI ^3^	F (%) M (%) *p* CI	F (%) M (%) *p* CI	F (%) M (%) *p* CI	F (%) M (%) *p* CI
Puebla 20/10		2 (10%) 1 (10%) 0.970 (0.1–32.6)	0 ---	6 (30%) 4 (40%) 0.608 (0.1–3.5)	4 (20%) 1 (10%) 0.558 (0.2–61.5)	0 2 (20%) 0.103 (0–1.7)
Tlaxcala 18/12		1 (5.6%) 2 (16.7%) 0.401 (0.01–4.7)	0 ---	4 (22.2%) 5 (41.7%) 0.293 (0.1–2.1)	3 (16.7%) 2 (16.7%) 0.986 (0.1–9.7)	2 (11.1%) 0 0.352 (0–5.2)
Hidalgo 23/7		2 (8.7%) 0 0.582 (0–11.8)	4 (17.4%) 0 0.3231 (0–3.7)	6 (26.1%) 4 (5.7%) 0.169 (0.04–1.7)	2 (8.7%) 6 (8.6%) 0.0001 (0.001–0.2)	0 1 (1.4%) 0.233 (0–5.8)
P + T + H ^4^ 61/29		5 (8.2%) 3 (10.3%) 0.735 (0.2–4.2)	4 (6.6%) 0 0.2042 (0–2.3)	16 (26.2%) 13 (44.8%) 0.088 (0.2–1.1)	9 (14.7%) 9 (31%) 0.087 (0.1–1.1)	2 (2.9%) 3 (10.3%) 0.223 (0.03–2.1)
Total	90	8 (8.9%)	4 (4.4%)	29 (32.2%)	18 (20%)	5 (5.5%)

^1^ Female/Male; ^2^ Two tails Mid-P exact (confidence limits); ^3^ Confidence Interval; ^4^ Overall ranch data.

## Data Availability

The data presented in this study are available on request from the first and corresponding authors.

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
