# Peer review of "Helminths of the Wild Boar (Sus scrofa) from Units of Conservation Management and Sustainable Use of Wildlife Installed in the Eastern Economic Region of Mexico"

_animals, 2021, doi:10.3390/ani11010098_

Round 1
Reviewer 1 Report
The subject is interesting, new and a first approach to the parasitic incidence in wild boars, in Mexico. As a general comment, the article will benefit from a revision of the structure (different line spacing between paragraphs, incorrect ciitation form and minor errors) and a proper scientific writing. The introduction and objectives should be further elaborated, including the study relevance (practically absent) of the concrete impact on Public Health.
The methodology should be described more clearly and another statistical methodology, however, could have valued the work. The results should be more explicit, namely when it presents highlighted and significant values ​​in bold, without any comment in the text and, obviously, deserve particular analysis, possible explanation and comments.
Comparability, made in the discussion, with several wild boars studies in other regions, should me more emphasized taking into account their diversity of results and possible breed, region or environmental influences. On the other hand, a special detail of the epidemiological study impact as well as its relevance in public health policies (and proposals) can and should be considered. The conclusion may be reinforced, considering the need of regulation, evidently based on more studies.
I congratulate the authors, in particular for the originality and relevance of the topic and suggest some changes in order to enhance the article.
Author Response
Response to Reviewer 1 Comments
Point 1: The subject is interesting, new and a first approach to the parasitic incidence in wild boars, in Mexico. As a general comment, the article will benefit from a revision of the structure (different line spacing between paragraphs, incorrect citation form and minor errors) and a proper scientific writing. The introduction and objectives should be further elaborated, including the study relevance (practically absent) of the concrete impact on Public Health.
Response 1: Based on the excellent comments of the reviewer, the general structure of the manuscript was reviewed. The spacing between paragraphs were revisited and corrected. The correct form of citation was reviewed and corrected. In general, the text was revised, and the appropriate formatting and wording adjustments were made. Emphasis about the importance of our work was also placed in the abstract, introduction, objective and conclusions sections.
Point 2: The methodology should be described more clearly and another statistical methodology, however, could have valued the work. The results should be more explicit, namely when it presents highlighted and significant values ​​in bold, without any comment in the text and, obviously, deserve particular analysis, possible explanation and comments.
Response 2: According to the reviewer's comments, the wording of the methodology was revised and the appropriate changes were made to avoid confusion; in fact, the stool examination and statistics sections were substantially improved. In the case of the results section, we in described the text the importance of our results.
Please, see in line 171, section “3.1 Determination of grastoenteric parasites”, the following text: “… During the overall analysis of data obtained with the 90 stool samples, we observed that Strongyloides and Oesophagostomum eggs were associated with the female sex of wild boar (p <0.05; Mid-P exact); however, during the analysis by ranch, this association only held for Strongyloides at Hidalgo ranch. In contrast, in the same Hidalgo ranch, Oesophagostomum was more frequent in males than in females (p <0.05; Mid-P exact).”.
Please, see in lines 324-331, the follow discussion. “Results here reported agree with data previously shown by other authors; however, we found in general, that female host were associated with Strongyloides and Oesophagostomum eggs; but, in contrast, in the same Hidalgo ranch, Oesophagostomum was more frequent in males than in females. These data only confirm that is controversial to define whether the sex of the host has any influence in the establishment of parasites and therefore, studies on this topic must continue. There are some examples of host sex influence in the development of the parasite, Taenia crassciceps, for example, the parasite strain is propagated in female mice to ensure the success of the procedure; but in other cases, as in human trichomoniosis, females host are marked associated with clinical development.
Please, see in line 41, section “Abstract”, the follow text were added: “The eggs of Strongyloides and Oesophagostomum were associated to female hosts”
Point 3: Comparability, made in the discussion, with several wild boars studies in other regions, should me more emphasized taking into account their diversity of results and possible breed, region or environmental influences. On the other hand, a special detail of the epidemiological study impact as well as its relevance in public health policies (and proposals) can and should be considered. The conclusion may be reinforced, considering the need of regulation, evidently based on more studies.
Response 3: In section "4. Discussion", we improved the discussion on studies of wild boar in other regions of the world. We took into account the reviewer's suggestion to focus on the influence of race, region, and environmental influence on the biodiversity of helminth species.
Please, see in line 314 the following text: “Thus, the wild board becomes a potential transmitter of parasites to humans or domestic animals, without taking into account the capacity of the wild boar as a reservoir for a great biodiversity of helminths. However, the biodiversity of helminths in wild boar and feral pigs is associated with the presence of abiotic factors, such as constant humidity and the absence of extreme temperatures that, combined with the presence of vectors and other hosts, can cause helminths conclude its life cycle. Additionally, it has been discussed whether there are strains more susceptible or resistant to parasites. Although the topic is still under discussion, it is known that the success of a parasite to establish itself in a host depends largely on the ability of the host itself to mount an adequate immune response. The issue can become controversial, because much has been discussed whether the immune response associated with helminths actually kills them or helps them survive.
Please, see in line 287 the added text about the discussion of the epidemiological study impact as well as its relevance in public health policies and the necessity of more studies: “At present, it is known that wild boar in Mexico are limited to hunting ranches; however, it is unknown how many of them have become feral animals. Additionally, the number of domestic pigs that have become feral animals is not known, and much less is the population of pig-wild boar hybrids known. There have been several attempts at the national level to determine the number of these swine in the country. In the states from the northern and central economic regions of Mexico, swine are considered invasive species. At the moment, there are no systematic data about the prevalence and distribution of wild board, which are also influenced by eating habits, forced migrations and human activity, who see these animals as an opportunity for profit, simply by hunting them without registration of health or economic activity. In this way, it is desirable that the real size of the swine feral animal population be known and its impact as an invasive species and transmitter-reservoir of pathogens verified.”
Please, see in line 375, section “5. Conclusions”, that que we reinforce the conclusion: “In conclusion, this is the first report that identifies the wild boar helminths of Mexico. Taking into account that the demand for wild boar meat consumption in Mexico is underestimated, more studies should be carried out to determine the impact of the transmission of these parasites to humans and animals destined for consumption and, if pertinent, accordingly. conducive studies, should strengthen sanitary regulations. Finally, in this work the potential risk of wild boar as a transmitter of diseases with importance for public health is identified.”
Point 4: I congratulate the authors, in particular for the originality and relevance of the topic and suggest some changes in order to enhance the article.
Response 4: On the contrary, we thank the reviewer for his time and thoughtful comments that helped improve our work.

Reviewer 2 Report
It is an interesting contribution to wild boar parasites in selected areas of Mexico. In my view, the data are presented understandably; results/conclusions are sound. Only some imperfections need to be corrected/considered by the authors:
l. 59-60: I would discriminate final, intermediate, and paratenic hosts of helminths.
l. 69: … enzyme-linked immunosorbent assays (ELISA) and Western blot, …
l. 95: … samples of 23 females of wild boars …
l. 113: Could the authors present the reason for multiplication by 20? What is the final calculation/value?
l. 125-126: Have the authors used any cultivation procedure to get excretory-secretory products?
l. 140, 150: What is the animal origin of anti-pig IgG?
l. 143: Absorbency should be replaced by absorbance.
l. 147-148: Briefly, a total of 300 μg of antigen? was separated on 73 mm gel (equivalent to 4 μg / mm of antigen?) of antigen (= remove?) on 11% …..
l. 168: Stages should replace structures.
l. 173: No significant difference was observed …. by sex … What is, therefore, meant by the bold values in Table 1?
l. 179, 183: … a cuticle >5 μm wide … Is this value valid if the body width is about 35 μm in total?
l. 181: Could the authors explain the reason for morphological comparison of the larva found with Trichinella larvae? Is there any conclusion on that comparison?
l. 246: …, while in the case of Taenia, the serum of a mouse infected experimentally was used.
l. 270-275: It is not necessary to repeat the results. Or, is there anything to be highlighted?
l. 308: Please, replace acanthocephalus with acanthocephalans.
l. 325: The value of 3% appears twice in the sentence. The frequency of 5% should be replaced by 5,5%.
There are some spelling/typing errors throughout the text to be corrected.
Author Response
Response to Reviewer 2 Comments
It is an interesting contribution to wild boar parasites in selected areas of Mexico. In my view, the data are presented understandably; results/conclusions are sound. Only some imperfections need to be corrected/considered by the authors:
Point 1: l. 59-60: I would discriminate final, intermediate, and paratenic hosts of helminths.
Response 1 We thanks to the reviewer for the time and the valuable comments that helped improve our work.
Please, see in line 61-65, the following added text “Wild boars are host of different stages of development of helminths with public health and veterinary importance. For example, wild boards and swine, in general, host the adult worms of Ascaris suum and larvae of Taenia solium, Echinococcus granulosus and Trichinella spiralis; indeed, wild boards can act as paratenic and accidental host of Toxocara and Fasciola, respectively”
Point 2: l. 69: … enzyme-linked immunosorbent assays (ELISA) and Western blot, …
Response 2: The change was done, please, see line 74
Point 3: l. 95: … samples of 23 females of wild boars …
Response 3: The sentence was modified in accordance with the observations of the reviewer. Please, see line 97
Point 4: l. 113: Could the authors present the reason for multiplication by 20? What is the final calculation/value?
Response 4: There was an error during the edition of the manuscript. Now the manuscript was corrected
Please, see line 112, section “2.3 stool examination”: “To determine the number of eggs per gram of feces, the number of eggs within the grids of each chamber was counted, the eggs outside the boxes were not taken into account, the total of eggs from the two chambers was added and multiplied by fifty”. Please see
Point 5: l. 125-126: Have the authors used any cultivation procedure to get excretory-secretory products?
Response 5: To obtain the excretion and secretion products, Trichinella larvae were cultured according to the procedure described in reference 16. To include the next paragraph in text, the references had to be reordered in the manuscript.
Please, see lines 124-128, section “2.5 antigens”: “In the case of Trichinella spiralis, the excretion and secretion products of the muscle larvae, strain MSUS/ME/92/CM-92, were prepared as previously reported [16]. Briefly, the larvae were isolated by the artificial digestion of meat obtained of rats experimentally infected and then incubated in RPMI 1640 medium (Gibco BRL, Grand Island, NY) at 37°C for 48 h in a humid atmosphere of 95% air and 5% CO2."
Point 6: l. 140, 150: What is the animal origin of anti-pig IgG?
Response 4: Please, see in line 140, section “2.6 Antibody determination”, that the correct name of the conjugated was placed: “Anti-pig IgG (whole molecule) peroxidase antibody produced in rabbit”
Please, see in line 149, the correction made in text
Point 7: l. 143: Absorbency should be replaced by absorbance.
Response 7: Please, see in line 142 it was replaced Absorbency by absorbance
Point 8: l. 147-148: Briefly, a total of 300 μg of antigen? was separated on 73 mm gel (?) of antigen (= remove?) on 11%.
Response 8: Please see in line 146 that the sentence was corrected: “Briefly, 300 µg of antigen were separated under reducing conditions on an 11% SDS-PAGE mini-gel (equivalent to 4 μg/mm of antigen). Afterward…”
Point 9: l. 168: Stages should replace structures.
Response 9: Please, see in line 166, the change in text.
Point 10: l. 173: No significant difference was observed …. by sex … What is, therefore, meant by the bold values in Table 1?
Response 10: we sought to better describe, briefly commenting in the text, the importance of the results with a significant difference.
Please, see in line 170, section “3.1 Determination of grastoenteric parasites”, the following text: “… During the overall analysis of data obtained with the 90 stool samples, we observed that Strongyloides and Oesophagostomum eggs were associated with the female sex of wild boar (p <0.05; Mid-P exact); however, during the analysis by ranch, this association only held for Strongyloides at Hidalgo ranch. In contrast, in the same Hidalgo ranch, Oesophagostomum was more frequent in males than in females (p <0.05; Mid-P exact).”.
Please, see in lines 324-331, that data was also discussed: “Results here reported agree with data previously shown by other authors; however, we found in general, that female host were associated with Strongyloides and Oesophagostomum eggs; but, in contrast, in the same Hidalgo ranch, Oesophagostomum was more frequent in males than in females. These data only confirm that is controversial to define whether the sex of the host has any influence in the establishment of parasites and therefore, studies on this topic must continue. There are some examples of host sex influence in the development of the parasite, Taenia crassciceps, for example, the parasite strain is propagated in female mice to ensure the success of the procedure; but in other cases, as in human trichomoniosis, females host are marked associated with clinical development.
Also, please, in line 41, section “Abstract”, the following text: “The eggs of Strongyloides and Oesophagostomum were associated to female hosts.
Point 11: l. 179, 183: … a cuticle >5 μm wide … Is this value valid if the body width is about 35 μm in total?
Response 11: We thank the reviewer for the excellent observation on larval measurements. We return to the original photographs to rectify the measurements. The 5um is the sum of cuticle and muscle; so to avoid confusion, we rectify the value of the cuticle:
Please, see in lines 189 and 194, section “3.2 Artificial digestion”, the following corrections: "…2.01 um for the larva of wild boar and 1.54 +/- 0.074 um for the larva of Trichinella".
Point 12: l. 181: Could the authors explain the reason for morphological comparison of the larva found with Trichinella larvae? Is there any conclusion on that comparison?
Response 12: Please, see in line 364 section “4. Discussion”, the following text that discuss the reasons to compare morphologically the larva found with those of Trichinella:
“Considering that trichinellosis is a zoonosis that is transmitted by the consumption of pig meat with viable muscular Trichinella larvae, in this work, we carried out an intentional search for these larvae and, if we had found them, we would have tried to propagate the cycle of life in an experimental way to later determine genus and species. However, we only found one larva by the artificial digestion of wild boar meat, so we proceeded to morphologically compare the recovered larvae with those of Trichinella that we propagated in the laboratory. We know that Trichinella larvae are not the only ones that can establish in skeletal muscle; other larvae, between others, could be of Toxocara or Baylisascaris genus, so we could not determine the classification. This finding suggests that the prevalence of tissue nematodes is low and therefore opens up a possibility for future epidemiological studies.”
Point 13: l. 246: …, while in the case of Taenia, the serum of a mouse infected experimentally was used.
Response 13: As mentioned in the section "2.5 Antigens", the experimental murine model of Taenia crassiceps was used to determine the presence of antibodies against cestodes; previously it has been shown that there are conserved epitopes among the cestodes of the genera Taenia and Echinococcus. Please check reference 29.
Please, see in line 246, "Figure 2", that the the word Taenia was changed to T. crassiceps
to avoid confusion
Point 14: l. 270-275: It is not necessary to repeat the results. Or, is there anything to be highlighted?
Response 14: The duplicate results on line 270-275 were removed. Please, see lines 265-278.
Point 15: l. 308: Please, replace acanthocephalus with acanthocephalans.
Response 15: Acanthocephalus was replaced with acanthocephalans. Please, see line 312.
Point 16: l. 325: The value of 3% appears twice in the sentence. The frequency of 5% should be replaced by 5,5%.
Please, see in line 358, the text was modified and the replacement of 5 to 5.5 was done
Point 17: There are some spelling/typing errors throughout the text to be corrected.
Response 178: Based on the excellent comments of the reviewer, the general structure of the manuscript was reviewed and spelling/typing errors corrected

Reviewer 3 Report
In the current manuscript, the authors investigated the endoparasitic fauna of wild boars in Mexico using different techniques. This is the first such study in this country.
In my opinion, the study is interesting, properly conducted, and worth publication. Results of this survey can be useful for veterinary and food safety services.
I have a few minor comments and questions:
- If possible please provide in the manuscript more data on the size of the wild boar population in Mexico
- Lines 57, 285 – why authors write the word ‘Countries’ with a capital letter?
- Line 59 - the comma is missing in the sentence ‘Ascaris Taenia’
- Cited by authors methodology of stool examination it is not in English (Técnicas para el diagnóstico de parásitos con importancia en salud pública y veterinaria; México, 2015; pp. 78–128). Therefore, could the authors describe which part of the McMaster chamber (one gird or both girds of the chamber, or whole McMaster chamber) was observed during examination.
- Eggs of Ascaris, Toxocara, or Taenia were not observed in the faeces, while IgG antibodies against these parasites were detected using ELISA. Could this be due to IgG antibody cross-reactions between helminths? Did the authors assess the specificity of the used serological methods?
- Please consider to change ‘number of parasites’ in Table 1 to ‘number of detected EPG/OPG’
Author Response
Response to Reviewer 3 Comments
In the current manuscript, the authors investigated the endoparasitic fauna of wild boars in Mexico using different techniques. This is the first such study in this country.
In my opinion, the study is interesting, properly conducted, and worth publication. Results of this survey can be useful for veterinary and food safety services.
I have a few minor comments and questions:
Point 1: If possible, please provide in the manuscript more data on the size of the wild boar population in Mexico
The Ministry of the Environment and Natural Resources is in charge of managing the hunting Ranches or UMAs, which are properties of persons, whom voluntarily use them for sustainable use. Systematic data about number of wild boars in Mexico is scarce, this information was incorporated in line 51-56 and discussed in line 287: “At present, it is known that wild boar in Mexico are limited to hunting ranches; however, it is unknown how many of them have become feral animals. Additionally, the number of domestic pigs that have become feral animals is not known, and much less is the population of pig-wild boar hybrids known. There have been several attempts at the national level to determine the number of these swine in the country. In the states from the northern and central economic regions of Mexico, swine are considered invasive species. At the moment, there are no systematic data about the prevalence and distribution of wild board, which are also influenced by eating habits, forced migrations and human activity, who see these animals as an opportunity for profit, simply by hunting them without registration of health or economic activity. In this way, it is desirable that the real size of the swine feral animal population be known and its impact as an invasive species and transmitter-reservoir of pathogens verified.”
Point 2: Lines 57, 285 – why authors write the word ‘Countries’ with a capital letter?
Response 2: Please, see in lines 60 and 279, the change of the word Country by country
Point 3: Line 59 - the comma is missing in the sentence ‘Ascaris Taenia’
Response 3: Please, see in line 61-65, the following added text “Wild boars are host of different stages of development of helminths with public health and veterinary importance. For example, wild boards and swine, in general, host the adult worms of Ascaris suum and larvae of Taenia solium, Echinococcus granulosus and Trichinella spiralis; indeed, wild boards can act as paratenic and accidental host of Toxocara and Fasciola, respectively”
Point 4: Cited by authors methodology of stool examination it is not in English (Técnicas para el diagnóstico de parásitos con importancia en salud pública y veterinaria; México, 2015; pp. 78–128). Therefore, could the authors describe which part of the McMaster chamber (one gird or both girds of the chamber, or whole McMaster chamber) was observed during examination.
Response 4: Please see line 107, the section “2.3 Stool examination”. Substantial changes were made. Two-chamber Mac master cameras were used to carry out this study, to determine the number of eggs per gram of feces, the number of eggs within the grids of each chamber were counted, afterward multiplied by 50
Point 5: Eggs of Ascaris, Toxocara, or Taenia were not observed in the faeces, while IgG antibodies against these parasites were detected using ELISA. Could this be due to IgG antibody cross-reactions between helminths? Did the authors assess the specificity of the used serological methods?
Response 5: Please, see in line 332 section “4. Discussion”, the following text, where we explain the lack of correlation between antibodies and eggs: “In this study we did not find Ascaris, Toxocara or any cestode eggs in the feces of the wild boar, but we did find IgG antibodies. This may be associated to the fact that the detection of antibodies, in the absence of clinical data, are not indicative of an active infection, but of contact with the etiological agent; in addition, the oviposition of helminths is not a continuous event but rather it is intermittent. Therefore, the determination of antibodies and stool examination techniques should be considered as complementary in order to make a global interpretation. There are factors, such as the sensitivity of stool techniques and the presence of cross-reacting antibodies in the different groups of helminths, which, if not properly interpreted, could over- or underestimate the prevalence of etiological agents. In fact, in this study, we take advantage of the cross-rection antibodies that exist between the cestodes to use the vesicular fluid of Taenia crassiceps to estimate the prevalence of cestodes in wild boar.
Point 6: Please consider to change ‘number of parasites’ in Table 1 to ‘number of detected EPG/OPG’
Response 6: Please, see the change in Table 1
